# Increased LRG1 Levels in Overweight and Obese Adolescents and Its Association with Obesity Markers, Including Leptin, Chemerin, and High Sensitivity C-Reactive Protein

**DOI:** 10.3390/ijms23158564

**Published:** 2022-08-02

**Authors:** Rashed Alhammad, Mohamed Abu-Farha, Maha M. Hammad, Thangavel Alphonse Thanaraj, Arshad Channanath, Nada Alam-Eldin, Reem Al-Sabah, Lemia Shaban, Abdulrahman Alduraywish, Fahd Al-Mulla, Abdur Rahman, Jehad Abubaker

**Affiliations:** 1Department of Pharmacology, Faculty of Medicine, Kuwait University, Safat 13110, Kuwait; rashed.alhammad@ku.edu.kw; 2Biochemistry and Molecular Biology Department, Dasman Diabetes Institute, Kuwait City 15462, Kuwait; mohamed.abufarha@dasmaninstitute.org (M.A.-F.); maha.hammad@dasmaninstitute.org (M.M.H.); nada.alamaldin@dasmaninstitute.org (N.A.-E.); 3Genetics and Bioinformatics Department, Dasman Diabetes Institute, Kuwait City 15462, Kuwait; alphonse.thangavel@dasmaninstitute.org (T.A.T.); arshad.channanath@dasmaninstitute.org (A.C.); fahd.almulla@dasmaninstitute.org (F.A.-M.); 4Department of Community Medicine and Behavioural Sciences, Faculty of Medicine, Kuwait University, Safat 13110, Kuwait; reem1@hsc.edu.kw; 5Department of Food Science and Nutrition, College of Life Sciences, Kuwait University, Safat 13060, Kuwait; dr.lemiashaban@gmail.com; 6Department of Internal Medicine, College of Medicine, Jouf University, Sakaka 72388, Saudi Arabia; dr-aaad@ju.edu.sa

**Keywords:** adolescents, high sensitivity C-reactive protein, Interleukin 6, leucine-rich α-2 glycoprotein 1, obesity, TNF-α

## Abstract

Leucine-rich α-2 glycoprotein1 (LRG1) is a member of the leucine-rich repeat (LRR) family that is implicated in multiple diseases, including cancer, aging, and heart failure, as well as diabetes and obesity. LRG1 plays a key role in diet-induced hepatosteatosis and insulin resistance by mediating the crosstalk between adipocytes and hepatocytes. LRG1 also promotes hepatosteatosis by upregulating de novo lipogenesis in the liver and suppressing fatty acid β-oxidation. In this study, we investigated the association of LRG1 with obesity markers, including leptin and other adipokines in adolescents (11–14 years; n = 425). BMI-for-age classification based on WHO growth charts was used to define obesity. Plasma LRG1 was measured by ELISA, while other markers were measured by multiplexing assay. Median (IQR) of LRG1 levels was higher in obese (30 (25, 38) µg/mL) and overweight (30 (24, 39) µg/mL) adolescents, compared to normal-weight participants (27 (22, 35) µg/mL). The highest tertile of LRG1 had an OR [95% CI] of 2.55 [1.44, 4.53] for obesity. LRG1 was positively correlated to plasma levels of high sensitivity c-reactive protein (HsCRP) (ρ = 0.2), leptin (ρ = 0.2), and chemerin (ρ = 0.24) with *p* < 0.001. Additionally, it was positively associated with plasma level of IL6 (ρ = 0.17) and IL10 (ρ = 0.14) but not TNF-α. In conclusion, LRG1 levels are increased in obese adolescents and are associated with increased levels of adipogenic markers. These results suggest the usefulness of LRG1 as an early biomarker for obesity and its related pathologies in adolescents.

## 1. Introduction

Childhood obesity is one of the most serious health challenges worldwide. The prevalence of childhood obesity in the Arabian Gulf Countries (GCC) is extremely high [1]. This might be due to the high caloric intake regimen and decreased levels of physical activity. It has been reported that 45% of children in Kuwait (5–19 years old) are either obese or overweight [2,3], which exceeds the prevalence in neighboring countries [3]. The prevalence of childhood obesity is on the rise globally. As obesity is associated with chronic diseases, like diabetes and several cardiovascular diseases (CVDs) [4,5], it is vital to develop therapeutic strategies by investigating the molecular pathways that contribute to the development of childhood obesity and identifying potential biomarkers that are helpful in diagnosis and progression of these conditions.

Leucine-rich α-2 glycoprotein 1 (LRG1) is a member of the highly conserved leucine-rich repeat (LRR) family [6], which plays several physiological functions related to cell adhesion, signal transduction, and protein-protein interaction [7,8]. In addition to its physiological functions, LRG1 has been reported to participate in several pathological conditions, including cancer and diabetes [9,10]. LRG1 has been shown to mediate cancer progression by mediating angiogenesis, promoting epithelial-mesenchymal transition (EMT), and inhibiting apoptosis [11,12].

Elevated plasma LRG1 levels were observed in patients with proliferative diabetic retinopathy (PDR) when compared to non-diabetic controls, indicating that it might be involved in the development of pathogenic ocular neovascularization [13]. Moreover, higher LRG1 levels were observed in young diabetes mellitus type 1 patients when compared to non-diabetic controls [14]. LRG1 has been linked to mediating insulin resistance, possibly by downregulating the expression of insulin receptor substrate 1 and 2 (IRS1 and IRS2) in hepatocytes [15]. LRG1 has been proposed to promote hepatosteatosis by upregulating de novo lipogenesis in the liver and suppressing fatty acid β-oxidation [15]. However, the exact mechanism remains unclear. Recently, it has been demonstrated that LRG1 can contribute to the development of metabolic dysfunctions caused by a high-fat diet (HFD) [15]. Specifically, it was shown that the elevated levels of LRG1 in circulation, which are produced by adipocytes, can interfere with or compromise the function of hepatocytes and hence participate in the development of insulin resistance and hepatosteatosis. Additionally, elevated levels of LRG1 were observed in fat depots of adults with obesity and positively correlated with waist circumference and body mass index (BMI) [10].

Overall, these previous reports indicate that LRG1 is implicated in the development of diabetes and obesity-related complications in adults. However, the role of LRG1 in these metabolic pathologies in children and adolescents has not been well-established. This study aimed to assess the role of LRG1 as a potential biomarker for childhood obesity. We investigated the association between LRG1 levels and obesity and CVD risk factors. We also aimed to compare LRG1 levels between obese, overweight, and normal-weight adolescents.

## 2. Results

### 2.1. Description of the Study Group and Obesity in School Adolescents

The demographic characteristics of the study group are summarized in Table 1. The participants were classified based on their BMI-for-age z-scores into 193 (45.4%) normal-weight, 89 (20.9%) overweight and 143 (33.7%) obese. Sex did not affect the classification since there was no significant difference in the prevalence of obesity or overweight between males and females (*p* = 0.944). The mean (SD) age of the participants was 12.3 (0.8) years, and 197 (45.7%) were males.

### 2.2. Plasma LRG1 Levels Are Significantly Elevated in Overweight and Obese Children

We found a significant difference in circulating LRG1 levels between obese and normal-weight adolescents, as well as between overweight and normal-weight participants (*p* = 0.007). The median (IQR) level of LRG1 was 27 (22, 35) µg/mL in normal-weight participants compared to 30 (24, 39) µg/mL in overweight and 30 (25, 38) µg/mL in obese (Figure 1). However, levels of LRG1 did not differ between overweight and obese groups (*p* > 0.05). Multinomial logistic regression showed that the odds of obesity and overweight were significantly associated with LRG1 in univariable analysis (*p* = 0.005 and 0.055, respectively) and after adjusting for potential confounders (*p* = 0.004 and 0.043, respectively) (Table 2). This was further highlighted when LRG1 was categorized into tertiles, where the upper tertile had an OR [95% CI] of 2.07 [1.12–3.8] for being overweight and 2.53 [1.42–4.50] for being obese, as compared to the lower tertile [reference] *p* = 0.002; Table 2. The association remained significant in the model adjusted for age and sex with AOR [95% CI] of 2.04 [1.08, 3.83] for being overweight and 2.64 [1.47, 4.74] for being obese (Table 2).

### 2.3. LRG1 Correlates with Obesity and Inflammatory Markers

We found a positive correlation between LRG1 and high sensitivity c-reactive protein (HsCRP) levels (Spearman’s rho = 0.2, *p* < 0.001; Figure 2A), leptin levels (Spearman’s rho = 0.2, *p* < 0.001; Figure 2B), and chemerin levels (Spearman’s rho = 0.24, *p* < 0.001; Figure 2C). Furthermore, circulating LRG1 levels were also positively correlated with IL6 (Spearman’s rho = 0.17, *p* < 0.001; Figure 3A) and IL10 (Spearman’s rho = 0.14, *p* = 0.003; Figure 3B). On the other hand, no correlation was reported between LRG1 levels and TNF-α (Spearman’s rho = −0.061, *p* > 0.05; Figure 3C).

## 3. Discussion

Obesity is a major health challenge that contributes to the development of diabetes and cardiovascular diseases. It has been reported that elevated levels of LRG1 were observed in young diabetic and obese adults compared to healthy adolescents and non-obese adults [10,14]. Therefore, it was crucial to investigate the association between LRG1 and several obesity markers and to compare LRG1 levels between obese, overweight, and normal-weight adolescents. In this report, we investigated the association between LRG1 and childhood obesity. Additionally, we assessed the association between LRG1 and several obesity markers, including HsCRP, chemerin, and leptin.

Our data showed significantly higher levels of plasma LRG1 in obese and overweight adolescents compared to normal-weight adolescents. Multinomial regression analysis revealed that adolescents with higher levels of plasma LRG1 were 2.5 times more likely to be obese after adjusting for multiple potential confounders suggesting that LRG1 might be an early obesity marker as the difference in plasma LRG1 levels between overweight and obese adolescents was not significant.

Our findings agree with several published reports showing that LRG1 correlates positively with obesity [10,15,16]. Higher serum LRG1 levels were observed in obese humans and mice compared to normal-weight humans and mice [15]. Another study showed that LRG1 correlates positively with waist circumference and BMI [10]. A positive association between LRG1 and BMI was also observed in patients with type-2 diabetes [16]. These similar findings indicate that ethnicity might not be involved in the association between LRG1 and obesity as the other studies were carried out on Caucasian and Asian participants while our study subjects were Arabs. The consistent positive association between LRG1 and obesity reported in diverse studies from various ethnic groups suggests that LRG1 could be used as a universal obesity marker.

In obesity, numerous inflammatory markers are upregulated, including HsCRP, which is involved in different diseases and detects low inflammatory processes [17,18]. Furthermore, we have previously published the HsCRP levels in the population from this current cohort and reported significant differences in the levels of HsCRP between every two groups (normal-weight vs. overweight, overweight vs. obese, and obese vs. normal-weight [19]. HsCRP is synthesized and secreted in hepatocytes in response to cytokines and inflammatory signals. It has been stated that high CRP levels associate positively with cardiovascular mortality in adults [20]. Moreover, CRP associates positively with an increased risk of insulin resistance and diabetes in adolescents and adults [21,22]. Our observations indicate that LRG1 and HsCRP might be part of a common pathway involved in the development of obesity as they both show a significant positive correlation with obesity.

IL6 regulates the synthesis and the secretion of CRP and promotes the development of inflammation [23]. It has been shown that higher IL6 levels were observed in obese patients and patients with chronic inflammatory conditions compared to healthy normal-weight subjects [24]. In addition, higher IL6 levels increase the risk of CVD, diabetes, and insulin resistance in obese patients [25]. On the other hand, IL10 plays a controversial role in humans, in which it exhibits pro- and anti- inflammatory effects. Some reports indicated that IL10 correlates negatively with obesity and diabetes [26,27,28]. In contrast, other studies indicated that IL10 correlates positively with obesity in young and adult females [29,30]. This suggests that IL10 correlation with obesity might depend on gender. Our findings show that LRG1 correlates positively with the inflammatory markers (IL6 and IL10), which could be due to the fact that IL6 and IL10 induce the activation of STAT3, an activator of LRG1 transcription [31,32].

Leptin is a well-established obesity marker that plays a major role in regulating immunity, appetite, and energy homeostasis [33]. Leptin promotes CRP expression and activates lymphocytes to secrete IL6 and IL10 through STAT3 and P38/MAPK signaling pathways [34,35]. Chemerin is an adipokine that is highly expressed in hepatocytes [36]. It plays a role in the chemoattraction of dendritic cells and macrophages during the immune response [37]. Moreover, chemerin was found to regulate glucose homeostasis, adipogenesis, and adipocyte metabolism [38]. Published reports showed that chemerin levels correlate positively with obesity and insulin resistance [39,40,41]. Similar to leptin, chemerin has also been shown to activate P38/MAPK signaling pathway resulting in an upregulation in IL6 [42]. Recently, LRG1 has been implicated in regulating P38/MAPK pathway as it has been shown to promote cancer metastasis and proliferation in thyroid and pancreatic cancers through P39/MAPK signaling pathway [43,44]. P38/MAPK signaling has been shown to play a role in the development of obesity as obesity correlates positively with increased P38/MAPK signaling [45,46].

Our findings suggest that LRG1 might be part of a common pathway involved in the development of obesity, possibly through P38/MAPK signaling pathway, as LRG1 has been shown to correlate positively with HsCRP, chemerin, leptin, IL6, and IL10 and that all of them have been implicated in regulating P38/MAPK signaling pathway. The regulation of LRG1 expression is not fully understood, however, it has been suggested that the IL6/STAT3 signaling pathway plays a major role in LRG1 transcription. Furthermore and in addition to IL6, many in vitro studies reported other inflammatory markers as possible regulators for LRG1 expression, including IL22, IL1β, IL17, TNF-α, IL4, IL33, IL10, and Oncostatin M [12,47,48,49,50]. These previous studies, together with the findings in our report, suggest that LRG1 is activated under different inflammatory conditions and imply that it acts downstream the inflammatory pathways. Hence, we can speculate that the increase in circulating LRG1 levels is the effect, and not the cause, of the increase in the inflammatory cytokines (leptin, HsCRP, IL6, and IL10). However, further studies should be carried out to explore the exact pathways involved in mediating LRG1 effects.

One of the strengths of our study is that it is one of the first studies to explore the association between LRG1 and obesity in adolescents as the majority of the previous studies were carried out on adults. This is important because establishing this association in a relatively healthy young population guarantees its independence from the complications commonly observed with adulthood obesity. Moreover, our data were adjusted for many potential confounding factors, and our study was carried out on reasonably large sample size. Lastly, numerous statistical tools were employed to robustly show the association between LRG1 and adolescent obesity. The main limitation of the study is the cross-sectional design which prevented us from establishing causality. Therefore, we plan to perform a longitudinal study to further explore the role of LRG1 in the development of obesity and the mechanism of its association with these inflammatory markers.

## 4. Materials and Methods

### 4.1. Study Participants, Ethics, Consent, and Permissions

Participants were recruited for this cross-sectional study from selected public middle schools in the State of Kuwait as previously described [51,52]. Ethical approval was obtained from The Ethics Committee of the Ministry of Health, Kuwait (No: 2015/248), The Ethics Committee of the Health Sciences Centre, Kuwait University (No: DR/EC/2338), and the Ethical Review Committee of Dasman Diabetes Institute (RA2017-026). The parents of all participants provided informed written consent, and verbal assent was obtained from all the study subjects before their enrolment in the study. Socio-demographic data were obtained through a self-administered questionnaire for the parents and a face-to-face interview with the participants.

### 4.2. Blood Collection and Biochemical Analyses

Five-millimeter venous blood was collected from each participant. Plasma was separated and stored at −80 °C till analysis. Glucose, complete blood count, iron profile, and 25-hydroxyvitamin D were assayed and reported. All the blood tests were carried out in a major tertiary care hospital following strict quality control measures.

### 4.3. ELISA Assays for LRG1 and HsCRP

Plasma LRG1 concentrations were determined using ELISA kit (Cat. # 27769; IBL, Gunma, Japan) with optimal dilution 1:4000 and HsCRP concentrations were determined using ELISA kit (Cat. # HK369; Hycult Biotech, North Brabant, Netherlands) with optimal dilution 1:1000. All assays were performed following the manufacturer’s instructions.

### 4.4. Multiplexing Assays

Leptin, chemerin, IL6, IL10, and TNF-α were assessed using a multiplexing immunobead array platform according to the manufacturer’s instructions (R & D Systems, Minneapolis, MN, USA). Median fluorescence intensities were collected on a Bioplex-200 system, and data were processed using the Bio-Plex Manager Software version 6 (Bio-Rad, Hercules, CA, USA), with five-parametric curve fitting.

### 4.5. Anthropometric Measurements

Standing height and bodyweight of the study participants were measured in a standardized manner, using a digital weight and height scale (Detecto: Webb City, MO, USA), with the participants standing erect without shoes and wearing light clothes. WHO growth charts were used to calculate BMI-for-age z-scores. Obesity was defined as BMI-for-age ≥+3 Standard Deviation (SD), while overweight was defined as BMI-for-age >+2 SD and <+3 SD.

### 4.6. Statistical Analysis

All measurements are reported as median (interquartile range: IQR). The associations between LRG1 and markers of interest, including HsCRP, leptin, chemerin, IL6, IL10, and TNF-α, were depicted graphically, and Spearman correlation coefficient was calculated. Multinomial logistic regression was used to determine the association between LRG1 and overweight or obesity. First, crude odds ratios (OR) were calculated, then we adjusted for sex and age group (10 years, 12 years, 13 years) to calculate adjusted odds ratios (AOR). Separate analyses were conducted while fitting LRG1, first as a continuous variable and then as a categorical variable. We used the Wald test to evaluate the statistical significance of these analyses. Associations with *p* < 0.05 were deemed to be significant.

## 5. Conclusions

In conclusion, higher plasma LRG1 levels were observed in obese and overweight adolescents compared with their normal-weight counterparts, indicating that LRG1 was associated with increased obesity risk. Our findings also suggest that LRG1 might be an early obesity marker as the difference in LRG1 levels between overweight and obese groups was not significant. Moreover, a positive correlation was observed between LRG1 and other obesity markers, including HsCRP, leptin, and chemerin. These findings shed light on the importance of LRG1 in adolescent obesity. In addition, the results indicate that LRG1 could be used as a potential prognostic and diagnostic tool in adolescent obesity. Lastly, our findings suggest that LRG1, chemerin, leptin, and HsCRP might be part of a common pathway involved in the development of obesity, possibly through P38/MAPK signaling pathway.

## Figures and Tables

**Figure 1 ijms-23-08564-f001:**
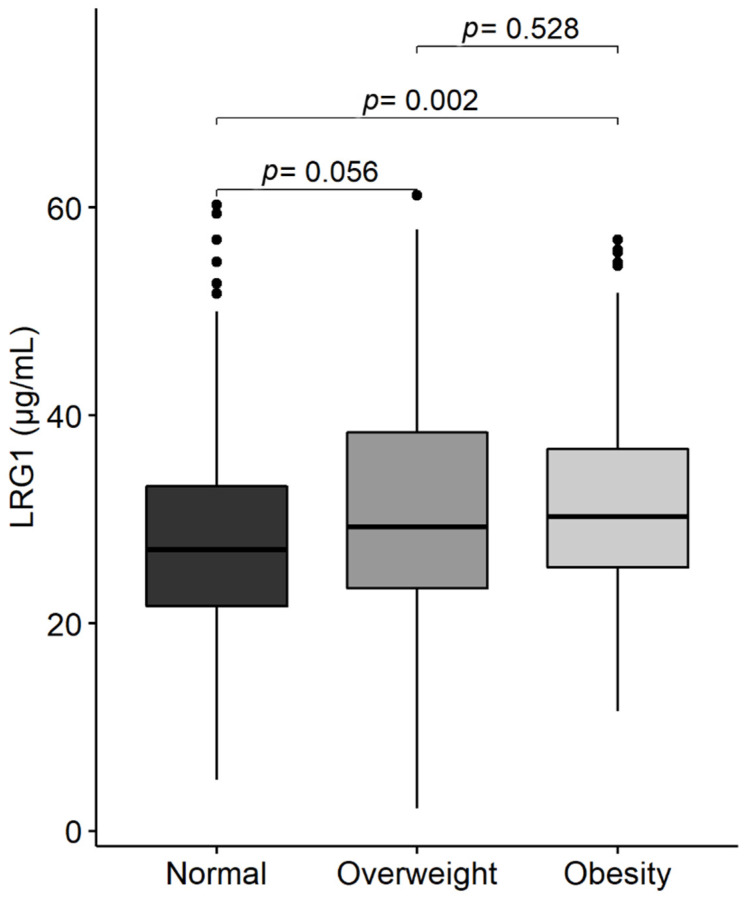
Distribution of LRG1 plasma level in normal-weight, overweight, and obese adolescents.

**Figure 2 ijms-23-08564-f002:**
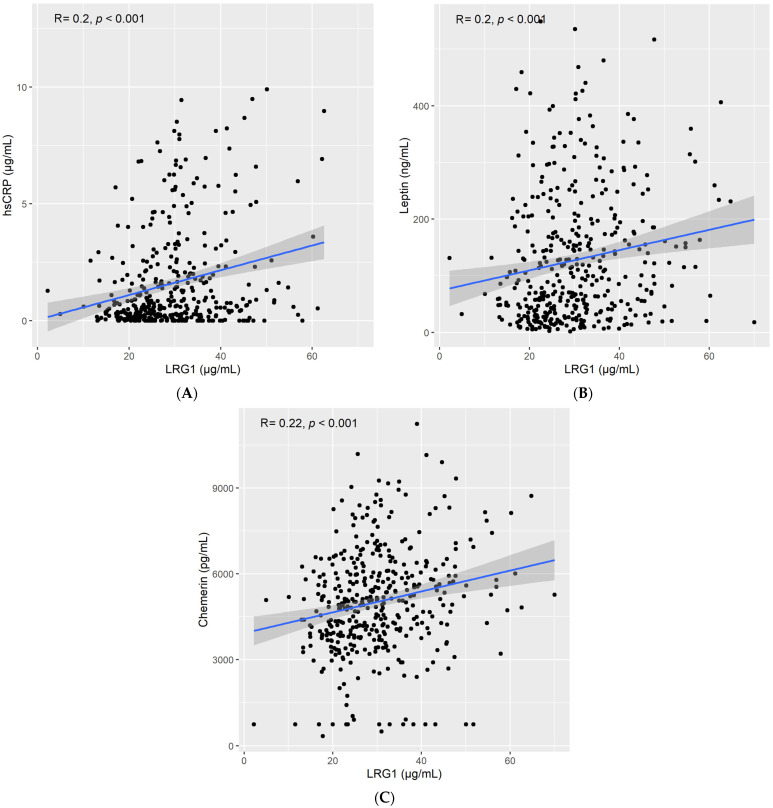
Correlation between LRG1 and (**A**) high sensitivity C-reactive protein (hsCRP), (**B**) leptin, and (**C**) chemerin.

**Figure 3 ijms-23-08564-f003:**
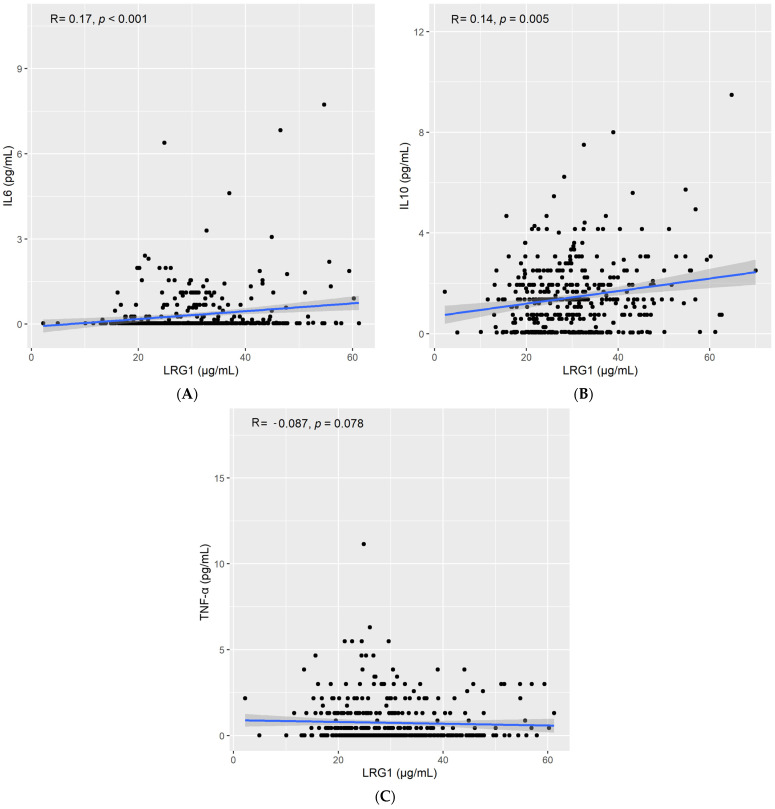
Correlation between LRG1 and (**A**) IL6, (**B**) IL10, and (**C**) TNF-α.

**Table 1 ijms-23-08564-t001:** Characteristics of 425 adolescents enrolled in the study.

Characteristic	Normal-Weight N = 193 ^1^	Overweight N = 89 ^1^	Obesity N = 143 ^1^	*p*-Value ^2^
Sex				>0.9
Female	107 (55%)	49 (55%)	76 (53%)	
Male	86 (45%)	40 (45%)	67 (47%)	
Age Group				0.8
10–<12 years	82 (42%)	37 (42%)	62 (43%)	
12–<13 years	67 (35%)	28 (31%)	52 (36%)	
13+ years	44 (23%)	24 (27%)	29 (20%)	
Glucose (mmol/L)	4.87 (0.72)	4.87 (0.74)	5.22 (1.76)	0.11
WBC (×10^9^ cells/L)	6.32 (1.61)	6.95 (2.03)	7.72 (1.90)	<0.001
RBC (×10^12^ cells/L)	4.99 (0.43)	5.04 (0.37)	5.11 (0.41)	0.011
Folate	1491 (282)	1449 (292)	1529 (319)	0.3
Vitamin D (nmol/L)	36 (23)	34 (20)	33 (19)	0.8

^1^ n (%); Mean (SD); ^2^ Pearson’s Chi-squared test; Fisher’s exact test; Kruskal–Wallis rank sum test. WBC, white blood cells; RBC, red blood cells.

**Table 2 ijms-23-08564-t002:** Association between overweight/obesity and LRG1 levels in univariable and multivariable multinomial logistic regression.

	Odds Ratio (OR) ^1^				Adjusted Odds Ratio (AOR) ^3^			
	Overweight		Obese		Overweight		Obese	
	OR ^1^ [95% CI] ^2^	*p*-Value	OR ^1^ [95% CI] ^2^	*p*-Value	AOR ^3^ [95% CI] ^2^	*p*-Value	AOR ^3^ [95% CI] ^2^	*p*-Value
LRG1 (univariable)	1.03 [1.00, 1.05]	0.055	1.03 [1.01, 1.06]	0.005	1.03 [1.00, 1.06]	0.043	1.04 [1.01, 1.06]	0.004
LRG1 (categories)								
Lower tertile (<24.8 µg/mL)	1 [Ref.]	-	1 [Ref.]	-	1 [Ref.]	-	1 [Ref.]	-
Middle tertile (>24.8 and <32.5 µg/mL)	1.22 [0.64, 2.32]	0.6	2.42 [1.38, 4.25]	0.002	1.20 [0.63, 2.30]	0.6	2.39 [1.36, 4.20]	0.002
Upper tertile (>35.5 µg/mL)	1.93 [1.04, 3.57]	0.037	2.55 [1.44, 4.53]	0.001	2.04 [1.08, 3.83]	0.027	2.64 [1.47, 4.74]	0.001

^1^ OR = Odds Ratio, ^2^ CI = Confidence Interval; ^3^ AOR: Adjusted odds ratio (adjusted for age, sex).

## Data Availability

The datasets used and/or analyzed during the current study are available from the corresponding author on reasonable request.

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
