# Peer review of "Increased LRG1 Levels in Overweight and Obese Adolescents and Its Association with Obesity Markers, Including Leptin, Chemerin, and High Sensitivity C-Reactive Protein"

_ijms, 2022, doi:10.3390/ijms23158564_

Round 1

Reviewer 1 Report

I have made some comments on the document. There are som minor langgauge changes. 

The graphs should have units on them, correlations are a bit low but still significant.

Author Response

I have made some comments on the document. There are som minor langgauge changes. 

Response: First, we would like to thank the reviewer for their feedback. We have fixed the marked comments in the revised version of the manuscript.

The graphs should have units on them, correlations are a bit low but still significant.

Response: Thank you for pointing this out. We have now added the missing units in all of the figures in the revised version of the manuscript.

Reviewer 2 Report

Current work show interesting results in relation to the association of  Leucine-rich α-2 glycoprotein1 (LRG1) with different parameters related to obesity and inflammation. The results are in agreement with other previous studies where it has been associated LRG1 with obesity in adults.

The results are of interest but as LRG1 have been previously associated to other metabolic parameters.-

Main concerns:

Association of LRG1 and obesity parameters are of interest but LRG1 has been previously associated with insulin resistance or other metabolic alterations. Have the authors explore the metabolic parameters in relation to LRG1? Please explains these findings.

  Minor comments:

Table 2 should be clarified for easier understanding od the analysis performed in each column.

Author Response

Current work show interesting results in relation to the association of Leucine-rich α-2 glycoprotein1 (LRG1) with different parameters related to obesity and inflammation. The results are in agreement with other previous studies where it has been associated LRG1 with obesity in adults.

The results are of interest but as LRG1 have been previously associated to other metabolic parameters.- 

Main concerns:

Association of LRG1 and obesity parameters are of interest but LRG1 has been previously associated with insulin resistance or other metabolic alterations. Have the authors explore the metabolic parameters in relation to LRG1? Please explains these findings.

Response: First, we would like to thank the reviewer for their feedback. Unfortunately, we don’t have information on metabolic parameters. The original cohort was designed to analyze the association between vitamin D levels and cognitive function, therefore, it was neither necessary nor logistically possible to obtain fasting blood samples required for insulin resistance and lipid profile analyses. We will make sure to account for such parameters in our future studies.

Minor comments:

Table 2 should be clarified for easier understanding od the analysis performed in each column.

Response: Thank you for pointing this out. We have now added some extra titles that we hope will clarify the analysis.

Round 2

Reviewer 2 Report

The present work in its present form is correct for publication. Unfortunately, some of the information that it could provide in relation to parameters of interest is not available.